

# Selective enrichment of active bacterial taxa in the *Microcystis* associated microbiome during colony growth

Carolina Croci[1,2], Gabriela Martínez de la Escalera[1,2], Carla Kruk[3,4], Angel Segura[3], Susana Deus Alvarez[1] and Claudia Piccini[1,2]

[1] Departamento de Microbiología, Instituto de Investigaciones Biológicas Clemente Estable, Montevideo, Uruguay
[2] Centro de Investigación en Ciencias Ambientales, Montevideo, Uruguay
[3] Departamento de Modelación Estadística de Datos e Inteligencia Artificial. Centro Universitario Regional del Este, Universidad de la República, Rocha, Uruguay
[4] Instituto de Ecología y Ciencias Ambientales, Sección Limnología, Universidad de la República, Montevideo, Uruguay

Corresponding author
Claudia Piccini,
cpiccini@iibce.edu.uy

## ABSTRACT

The toxic cyanobacterium *Microcystis* causes worldwide health concerns, being frequently found in freshwater and estuarine ecosystems. Under natural conditions, *Microcystis* spp. show a colonial lifestyle involving a phycosphere populated by a highly diverse associated microbiome. In a previous study, we have proposed that colony formation and growth may be achieved through mechanisms of multispecies bacterial biofilm formation. Starting with single-cells, specific bacteria would be recruited from the environment to attach and create a buoyant biofilm or colony. This progression from a few single cells to large colonies would encompass the growth of the *Microcystis* community and bloom formation. In order to test this, we applied 16S rDNA metabarcoding to evaluate the changes in bacterial community structure (gDNA) and its active portion (cDNA) between different sample sizes obtained from a *Microcystis* bloom. Bloom sample was sieved by size, from one or a few cells (U fraction) to large colonies (maximum linear dimension ≥ 150 μm; L fraction), including small (20–60 μm, S fraction) and medium size (60–150 μm, M fraction) colonies. We found that gDNA- and cDNA-based bacterial assemblages significantly differed mostly due to the presence of different taxa that became active among the different sizes. The compositional variations in the communities between the assessed sample sizes were mainly attributed to turnover. From U to M fractions the turnover was a result of selection processes, while between M and L fractions stochastic processes were likely responsible for the changes. The results suggest that colony formation and growth are a consequence of mechanisms accounting for recruitment and selection of specific bacterial groups, which activate or stop growing through the different phases of the biofilm formation. When the final phase (L fraction colonies) is reached the colonies start to disaggregate (bloom decay), few cells or single cells are released and they can start new biofilms when conditions are suitable (bloom development).

## INTRODUCTION

Among bloom-forming cyanobacteria, *Microcystis* species are the most frequent worldwide (*De Leon & Yunes, 2001*; *Zurawell et al., 2005*; *Huisman & Hulot, 2005*; *González-Piana et al., 2011*; *O'Neil et al., 2012*; *Paerl & Otten, 2013*; *Srivastava et al., 2013*; *Harke et al., 2016*). *Microcystis* blooms are of special concern because they are mainly composed by populations able to produce secondary metabolites called microcystins, which are toxic to animals and humans (*Vezie et al., 1998*). These blooms can be very dense in eutrophic ecosystems (*Kruk et al., 2023*; *Bonilla et al., 2023*), where high nutrient concentration promotes *Microcystis* growth. During a bloom, *Microcystis* can be found in a wide range of organism's sizes, from single cells (4 μm) to large colonies visible to the naked eye (*Kruk et al., 2017*; *Reynolds et al., 1981*). The size variability of *Microcystis* spp. has been related to physiological and ontogenetic stages driven by the environmental conditions (*Reynolds et al., 1981*; *Deus et al., 2020*). For example, the size distribution of *Microcystis* spp. biovolume appears as a good predictor of active toxin production, being the colonies in the 60–150 μm size fraction good indicators of higher toxicity (*Deus et al., 2020*; *Wang et al., 2013*). Moreover, *Gan et al. (2012)* showed that the addition of microcystin to *Microcystis* cultures provoked a significant increase in colony size.

The colonial lifestyle provides *Microcystis* with multiple benefits, including protection from grazing (*Becker, 2010*; *Stahl, 2017*; *Xiao et al., 2017*), adaptation to light variation (*Stahl, 2017*; *Xiao et al., 2017*), protection from chemical stressors (*Xiao et al., 2017*), increased water column transparency (*Harel et al., 2012*; *Stahl, 2017*), and a milieu where nutrients are efficiently recycled (*Stahl, 2017*; *Ma et al., 2014*). Yet, from an evolutionary point of view, *Microcystis* belongs to a clade of unicellular cyanobacteria (*Schirrmeister et al., 2013*), and it is well known that when isolated under laboratory conditions the colonies give usually place to unicellular cultures (*Xiao et al., 2017*; *Xiao, Li & Reynolds, 2018*). Dilution of samples to obtain an isolate will also likely dilute the *Microcystis*-associated and the free-living bacteria, thus preventing the conditions that favor aggregation of *Microcystis* cells and colony formation.

*Microcystis* colonies are embedded in a layer of mucilage (*Kehr & Dittmann, 2015*; *Liu, Huang & Qin, 2018*) populated by a high number of heterotrophic and autotrophic bacteria growing at expense of the extracellular polysaccharides (EPS) availability (Fig. 1). This EPS-rich space inhabited by a high diversity of microorganisms (the phycosphere) can have ecosystem-level effects on several processes, *e.g.*, nutrient cycling and toxin biosynthesis (*Bell & Mitchell, 1972*; *Seymour et al., 2017*). In the *Microcystis* phycosphere, the heterotrophic bacteria were found to stimulate the cyanobacterial growth, to induce the production of EPS (*Wang et al., 2015*), and to allow the cyanobacteria accessing specific compounds, *e.g.*, vitamins and outer membrane lipopolysaccharide, while providing bacteria with highly bioavailable carbon (*Xie et al., 2016*).

Recently, *Pérez-Carrascal et al. (2021)* have reported that *Microcystis* associated microbiome is genotype-specific and that closely related genotypes have similar microbiomes. Besides, co-cultivation of axenic, single-celled cultures of *Microcystis* with a natural *Microcystis*-associated heterotrophic bacteria isolated from *Microcystis* colonies
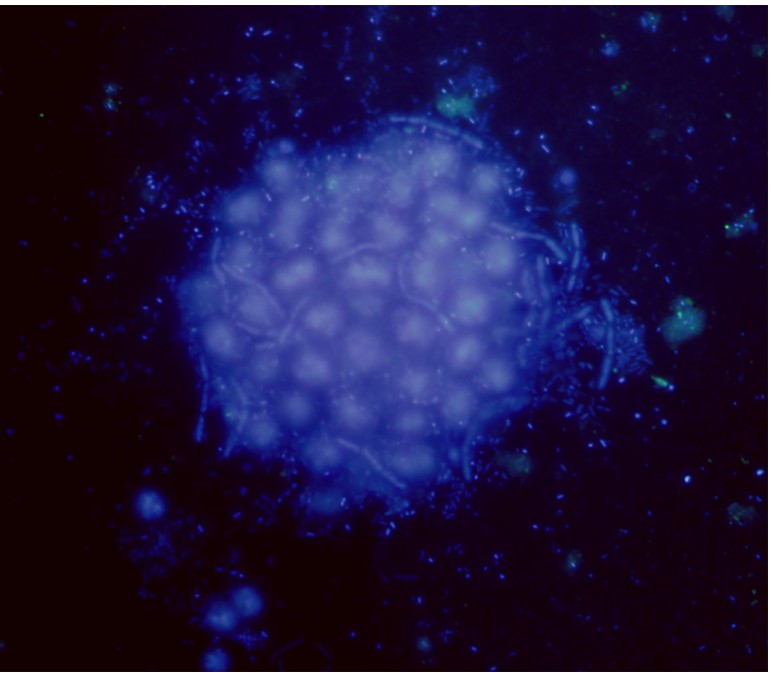

**Figure 1** ***Microcystis* phycosphere.** DAPI-stained *Microcystis* colony showing the bacteria attached to the mucilage.            

stimulated the production of EPS, allowing to reconstitute a colonial phase (*Shen et al., 2011*). On the other hand, removing the EPS has shown to have a detrimental effect on the auto-aggregation abilities of heterotrophic bacteria isolated from *Microcystis* colonies, suggesting a relevant role of EPS in the recruitment of bacteria by promoting their attachment (*Zhang et al., 2018*).

Taken together, current evidence points to the existence of a metabolic interdependence between *Microcystis* and its associated microbiome (*Jackrel et al., 2019*; *Cook et al., 2020*). Moreover, it has been suggested that in natural conditions the ability of *Microcystis* to compete with other phytoplankton groups could not be determined by the toxin production but by the interaction with its associated microbiome (*Schmidt et al., 2020*). In addition, the presence of quorum sensing (QS) mechanisms has been reported for *Microcystis*, which stimulate colonial growth *via* acylated homoserine lactones (AHL) generated either by *Microcystis* or by other bacterial sources (*Herrera & Echeverri, 2021*). Through this multispecies QS these organisms are able to form a three-dimensional biofilm structure (*Shi et al., 2022*) that functions as a complex holobiont (*Piccini et al., 2024*). This symbiotic relationship would have been established early in the evolution of this unicellular cyanobacterium and could be the key to the success of *Microcystis* in changing environments. Thus, understanding the mechanisms underlying colony formation in *Microcystis* would help to unveil the role of the associated microbiome in the evolution and environmental performance of this holobiont.

For stream biofilms, it has been described that assembly processes would be mainly deterministic (*Besemer et al., 2012*; *Veach et al., 2016*). As biofilms become thicker,
nutrient gradients increase, generating different microenvironments that can support a greater diversity of metabolic lifestyles, giving rise to additional selective pressures on the organisms (*Fowler et al., 2023*). Despite different studies examined community assembly processes in different kind of aquatic biofilms, the conclusions are often contrasting and the factors driving these different observations are not yet well understood or addressed (*Vignola et al., 2018*; *Matar et al., 2021*).

Considering the aforementioned background on the role of heterotrophic bacteria and microcystin in colony formation, we hypothesize that the development and growth of *Microcystis* colonies is driven by multispecies bacterial biofilm formation mechanisms. The formation and growth of these biofilms, or colonies, reflect the onset and development of the bloom. In this context, the increase in colony size is related to microbiome dynamics (growth and decay of different populations driven by different community assembly mechanisms) and microcystin production. Thus, our aim was to assess the changes in the bacterial community during the development and growth of *Microcystis* colonies and the underlying assembly mechanisms. To this end, we analyzed the microbiome associated to size-fractionated *Microcystis* colonies obtained from a bloom by 16S rRNA metabarcoding using genomic DNA (gDNA, considered as the whole bacterial community) and RNA (cDNA, representing the metabolically active fraction) (*Blazewicz et al., 2013*). Beta diversity among the bacterial communities associated with each sample size was also assessed to establish the community assembly mechanisms. To determine the potential link between toxin production and sample size, microcystin concentration was also assessed.

## MATERIALS AND METHODS

### Samples processing

A water sample was taken in January 2017 at Salto Grande reservoir (31°12′59.3″S 57°56′00.1″W) during a *Microcystis* spp. bloom. Salto Grande is a eutrophic-hypereutrophic reservoir that suffers recurrent cyanobacterial blooms of *Microcystis* spp. (*Bordet, Fontanarrosa & O'Farrell, 2017*; *Chalar, 2006*; *Kruk et al., 2017*; *Lepillanca et al., 2018*; *Martínez de la Escalera et al., 2017*, *2023*; *O'Farrell, Bordet & Chaparro, 2012*). Size-fractionated water samples were used to analyze the structure, composition and activity of the bacterial community associated with *Microcystis* colonies (for details see *Deus et al., 2020*). Briefly, 5 L surface water samples were taken using a sterile plastic bottle and then fractionated through nylon nets in sequential mesh sizes of 150, 60 and 20 μm. From each size-fractionated sample, a 15 mL aliquot was filtered through a 0.2 μm GTTP filter (Millipore). 250–300 ml of the water passing through all these sizes was finally filtered through a 0.2 μm GTTP filter (Millipore) and accounted for the unicellular fraction. All the filters were preserved in RNAlater (Thermo Fisher Scientific, Waltham, MA, USA) until nucleic acid extraction. This resulted in concentrated *Microcystis* size fractions, which were classified according to their maximum linear dimension (MLD) in large (L; >150 μm), medium (M; 150–60 μm), small (S; 60–20 μm) and unicellular (U; <20 μm) that includes single cells and colonies of a few cells (for details see *Deus et al., 2020*).

Environmental variables were measured *in situ* (pH, water temperature, turbidity) using a multiparameter probe (Horiba). Chlorophyll a concentration was determined by High-Performance Liquid Chromatography (HPLC) following the Standard Methods 10200 H (22nd edition).

## Size fraction estimation

In order to confirm the size classes, a 10 mL subsample from each size fraction was used to measure *Microcystis* individual dimensions using 1 mL Sedgwick-Rafter chambers under an inverted Olympus IX81 microscope at ×100 magnification. The maximum linear dimension (MLD, μm) of single unattached cells and colonies was determined. Width and depth of organisms were measured using the Cell F software (Olympus Cell Series). Individual cell and colony volume (μm$^3$) was calculated by approximating a prolate spheroid (*Alcántara et al., 2018*). Based on the obtained volume, the equivalent spherical diameter (ESD, μm$^3$) was calculated.

## DNA and RNA extraction

The gDNA extraction was performed according to the protocol described in *Deus et al. (2020)*. Briefly, the filter pieces and lysis extraction buffer were put in a 2 ml tube containing ceramic beads (2 mm diameter), homogenization during 40 s at 6 m/s in a FastPrep (MP Biomedicals), separation of nucleic acid with chloroform:isoamyl (24:1), nucleic acid precipitation with isopropanol and incubation for 1 h at room temperature (20 °C), and a washing with cold 70% (v/v) ethanol.

For the RNA extraction, the PureLinkTM RNA MiniKit kit (Thermo Fisher Scientific, Waltham, MA, USA) was used. Retrotranscription reactions were performed to obtain the cDNA using the High Capacity RNA-to-cDNA Kit (Thermo Fisher Scientific, Waltham, MA, USA) according to the manufacturer instructions.

## Toxic cell abundance and microcystin concentration

The abundance of microcystin-producing cells for each sample size was assessed through quantification of the *mcyE* gene by quantitative real-time PCR (qPCR) (*Deus et al., 2020*). Briefly, two microliters of DNA extracts from each fraction (ca. 50 ng DNA) were applied to the Power SYBR Green PCR (Invitrogen, Carlsbad, CA, USA) in a final reaction volume of 20 μL using *mcyE* primers (*Sipari et al., 2010*). Cycling conditions were 2 min at 50 °C, 15 min at 95 °C and 40 cycles of 15 s at 94 °C, 30 s at 60 °C and 30 s at 72 °C, including a last melting step from 65 to 95 °C at 1 °C steps each 4s. A 96 FLX Touch TM thermal cycler (Bio-Rad, Hercules, CA, USA) was used. Quantification curves were achieved using serial dilutions, from 1/10 to 1/100,000 of a vector containing the *mcyE* cloned gene and applied to qPCR in the same PCR plate, where the samples were assayed by triplicate (*Martínez de la Escalera et al., 2017*). Negative controls without DNA were included at each run.

To determine microcystin concentration present in each size fraction an indirect competitive Enzyme-Linked ImmunoSorbent Assay (ELISA) using rabbit polyclonal antibodies was performed according to *Pírez et al. (2013)*. Microcystin concentration was

normalized by the *mcyE* copy number per mL (qPCR data from *Deus et al., 2020*) and the result was interpreted as microcystin produced per cell.

## 16S rRNA sequencing

The V4 region of the 16S small subunit ribosomal RNA (SSU rRNA) gene was amplified from gDNA and cDNA using the universal par of primers 515f and 806r (*Parada, Needham & Fuhrman, 2016*) and amplicons were sequenced at the University of Minnesota on an Illumina MiSeq platform (2 × 300 bp).

## Sequence analysis

All the bioinformatic analysis were performed with the free software R, version 4.2.1 (*R Core Team, 2022*) and DADA2 version 1.16.0 (*Callahan et al., 2016a*, *2016b*). The raw sequences were provided already demultiplexed, which were processed to eliminate those showing poor quality according to the following criteria: sequence quality, sequence with ambiguous bases (N), chimeras and amplicon length. The resulting sequences were clustered into ASVs (Amplicon Sequence Variant, 100% sequence similarity), which were then assigned to their taxonomy by comparing with the SILVA SSU database version 138.1 (*Quast et al., 2013*; *Yilmaz et al., 2014*). After the generation of a phyloseq object (*McMurdie & Holmes, 2013*), ASVs were filtered removing those coming from chloroplasts, mitochondria and Eukarya, and then according to the prevalence and abundance of the ASVs. The 16S rRNA obtained sequences can be found at GenBank (PRJNA1169899).

## Data analysis

Rarefaction curves were generated with the vegan package (*Oksanen et al., 2022*) to verify that the sequencing depth was optimum and that all samples have had the same sampling effort. We normalized the data using proportions (*McKnight et al., 2019*). To explore differences among size fractions, α-diversity (excluding *Microcystis* ASVs) indexes were estimated: Richness, Shannon, Simpsons, and Equitability. Differences between the bacterial communities from different sample sizes (β-diversity) for gDNA- and cDNA-based samples were addressed using Bray-Curtis dissimilarity (also excluding *Microcystis* ASVs). To determine if differences in community composition between samples were statistically significant, we performed a PERMANOVA using the function adonis2 in the vegan package with 999 random permutations (*Oksanen et al., 2022*). Spearman correlation analyses between different sample sizes were performed using the ASVs abundance and composition obtained from each sample.

The ASV abundance table (abundance of each ASV per sample) was used to calculate the percentage corresponding to each phylum. For those phyla showing a relative abundance higher than 5% at least in one fraction, we calculated the percentage corresponding to each order, family or genus (depending on the depth at which they were classified) belonging to that phylum. Once the percentages were calculated, we looked for those reaching more than 5% in any fraction and calculated its shift in abundance from one fraction size to the next, both for gDNA and cDNA and between them. When the taxa were

analyzed at the genus level, those whose relative abundance was below 5% and those that were not assigned were excluded from the analysis.

To quantify the contribution of different ecological processes to the compositional variations of microbial communities (stochastically or deterministically assembled), a pairwise modified Raup-Crick β-diversity dissimilarity index ($RC_{(BC)}$) was calculated using Bray-Curtis (*Raup & Crick, 1979*) for the gDNA-based communities (*Chase et al., 2011*; *Stegen et al., 2013*). Modified $RC_{(BC)}$ were calculated using RC.pc from {iCAMP} in R (*Ning et al., 2020*). The modified Raup-Crick$_{(BC)}$ values were interpreted as follows: $RC_{(BC)}$ > 0.95 suggest that the compositional variations are favored by deterministic factors, $RC_{(BC)}$ < −0.95 were interpreted as significantly more similar as expected by chance (deterministic factors favoring similar microbes as the dominant process); while $RC_{(BC)} \leq$ 0.95 suggest that the observed and null communities are comparable and compositional variations resulted from stochastic processes (*Ma & Tu, 2022*).

In order to determine if the changes in community structure were due to species replacement (turnover) or loss (nestedness) we used the Sorensen index to calculate the overall beta diversity, the Simpson index for the turnover component and the nestedness component was the resultant fraction of Sorensen dissimilarity ($ß_{Sorensen} = ß_{Simpson} + ß_{Nestedness}$) ({betapart} package (*Baselga & Orme, 2012*)). Finally, to visualize the differential taxa between different sample sizes, we analyzed the relative abundances of each taxon (at the maximum level that could be identified, be it phylum, class, order, family or genus) found for each sample size.

## RESULTS

### Environmental conditions, *Microcystis* colony fractions and toxicity

Water temperature during sampling was 29.4 °C, pH 9.7, turbidity 62.8 NTU and chlorophyll-a concentration of 215.5 µg L$^{-1}$, reflecting the bloom conditions.

The size of the fractions obtained by sequential filtration using different mesh sizes was confirmed by microscopy (Table S1). The abundance of toxic *Microcystis* cells in each size fraction, assessed as *mcyE* copies mL$^{-1}$, was 1.5 for the U fraction, and $2.9 \times 10^3$, $6.2 \times 10^3$ and $6.4 \times 10^3$ for the S, M and L fractions, respectively. The highest microcystin concentration per toxin-producing cell was found in the M fraction (*p*-value = 0.09; Fig. 2).

### Bacterial diversity (gDNA) in the colony fractions

A total of 959,576 reads were obtained, after processing and quality filtering reduced to 652,748, which resulted in 1,732 ASVs. All samples reached the plateau in the rarefaction curve (Fig. S1). The highest ASV richness was found in the smallest fractions (U and S). A pattern showing decreased diversity towards the higher fractions was found (Fig. 3). No differences were found using Simpson and Equitability indexes (Table S2).

Bray-Curtis dissimilarity based PCoA showed differences between the communities obtained from gDNA and cDNA (PERMANOVA, *p*-value = 0.03). The first two axes explained 56.56% of total variance. The first axis separated the gDNA from cDNA samples (37.53%), while the second split the U fractions from the other size fractions (19.03%) (Fig. S2). Accordingly, the bacterial community composition obtained from gDNA and

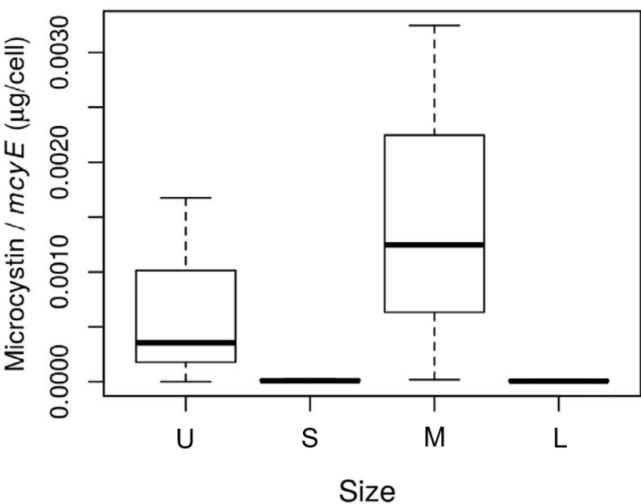

**Figure 2 Toxin concentration among size class in *Microcystis* cyanobacteria.** Microcystin concentration (µg per cell) measured for each sample size class: U (Unicelular, <20 µm), S (Small, 20–60 µm), M (Medium, 60–150 µm) and L (Large, >150 µm) fractions. Taken from *Deus et al. (2020)*.

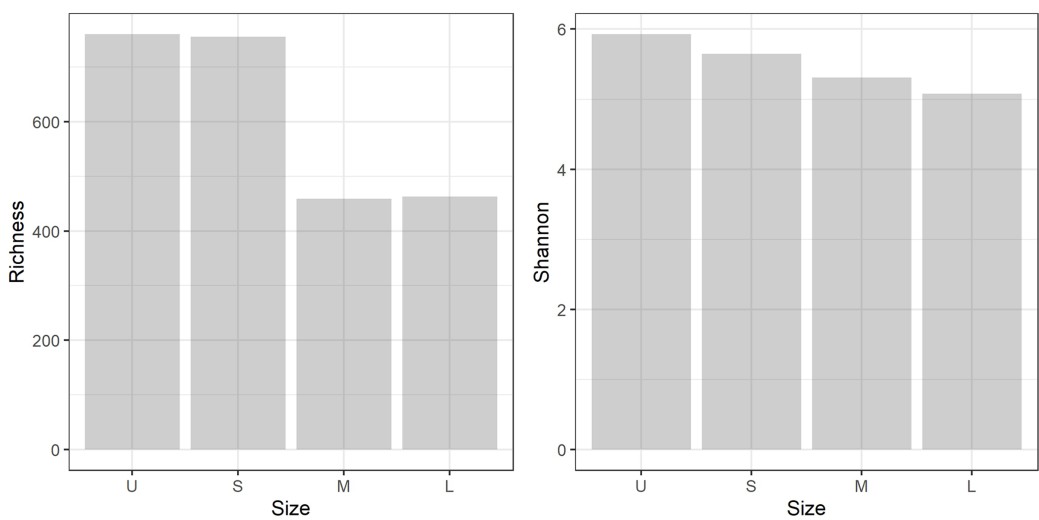

**Figure 3 Microbiome diversity.** Bacterial community richness (A) and Shannon diversity (B) obtained from the gDNA of the different sample size classes: U (Unicelular, <20 µm), S (Small, 20–60 µm), M (Medium, 60–150 µm) and L (Large, >150 µm) fractions.

cDNA for each size fraction showed low correlation values, as evidenced by the $r^2$ values obtained (0.39, 0.17, 0.21 and 0.23 for U, S, M and L fractions, respectively) (Fig. 4A). Positive, significant correlations were found between the bacterial community composition of M and L fractions for both gDNA and cDNA-based communities ($r^2$ 0.55 and 0.31, respectively) (Fig. 4A). A negative correlation value was only found between the
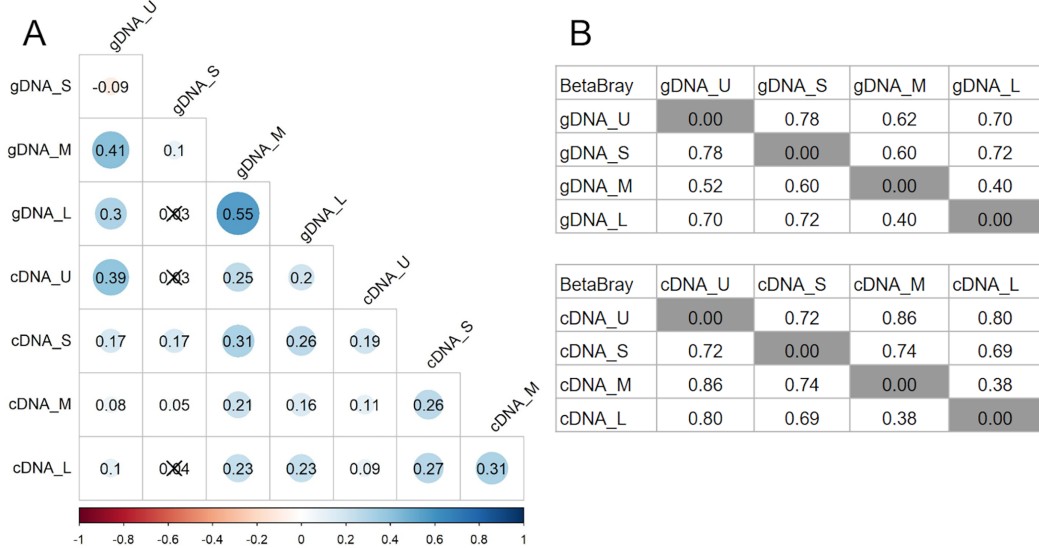

**Figure 4 Relationships between microbiomes of colonies of different sizes.** (A) Correlation matrix based on the ASVs abundances from each size fraction in DNA and cDNA-based samples. The Spearman $r^2$ values are shown. Non-significant correlations are crossed ($p$-value > 0.05). (B) Bray-Curtis dissimilarity matrices between gDNA and cDNA samples. U (Unicelular, <20 μm), S (Small, 20–60 μm), M (Medium, 60–150 μm) and L (Large, >150 μm) fractions.

community composition (gDNA) of U and S fractions, while the active community (cDNA) from these two fractions were weakly but positively correlated ($r^2$ = 0.19), indicating significant shifts in active taxa taking place when progressing from the unicellular stage to the next size (U to S). Overall, Bray-Curtis dissimilarity index between the bacterial communities from the smaller fractions (U and S) for both gDNA and cDNA-based samples was 78% and 72%, respectively (Fig. 4B). This index showed lower differences between M and L fractions for both gDNA and cDNA samples, with values of 40% and 38% respectively (Fig. 4B).

Pairwise comparisons across sample sizes for the gDNA-based samples showed mean positive $RC_{(BC)}$ values > 0.95, except between M and L fractions ($RC_{(BC)}$ = 0.204), indicating that changes in community composition from U to M sizes are primarily due to selection and from M to L mostly to drift.

## Size-specific bacterial community composition

*Microcystis* was the most abundant cyanobacterial genus detected through the whole dataset, with relative abundances ranging from 5% to 13% (Fig. S3). Excluding *Microcystis*, the most abundant phyla were Proteobacteria and Bacteroidetes, followed by Cyanobacteria, Planctomycetes and Actinobacteria (Fig. 5A, left panel). As the community progressed to larger colony sizes, the relative abundance of Proteobacteria increased (27% to 62%), while Bacteroidetes (23% to 10%), Cyanobacteria (12% to 4%), Planctomycetes (10% to 4%) and Actinobacteria (12% to 7%) decreased (Fig. 5A, left panel).

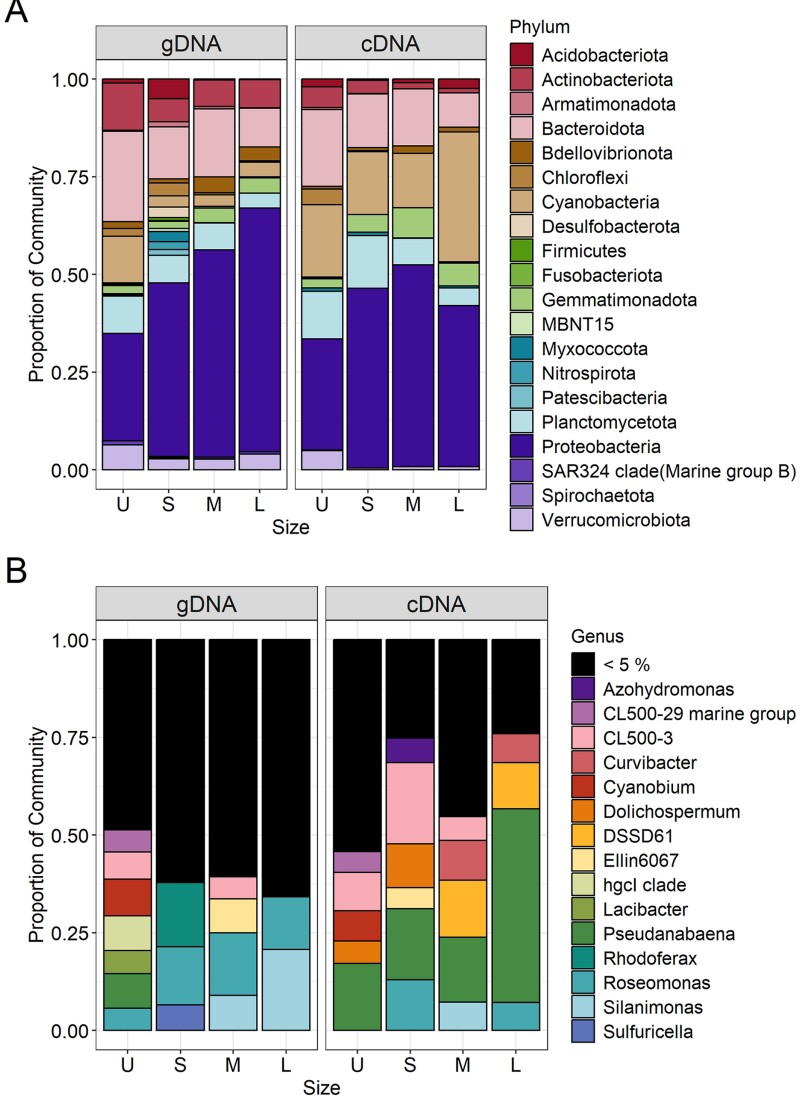

**Figure 5 Bacterial community composition (gDNA) and the active bacterial fraction (cDNA) at the phylum (A) and genus level (B) of the different sample size classes.** U (Unicelular, <20 μm), S (Small, 20–60 μm), M (Medium, 60–150 μm) and L (Large, >150 μm).

## U fraction

Due to the filtration method used to obtain the unicellular fraction, it can also include the free-living bacterial community. This fraction showed the lower representation of Alphaproteobacteria (14%) and Gammaproteobacteria (13%), with only the Burkholderiales order presenting an abundance greater than 5% (10%). In this fraction, a greater variety of Bacteroidetes orders were present (Chitinophagales, Flavobacteriales, Kapabacteriales and Sphingobacteriales), but with only Chitinophagales and Sphingobacteriales having values equal to or greater than 5% (9 and 6% respectively). In the case of Cyanobacteria, only *Cyanobium* and *Pseudanabaena* genera were present,

showing its highest abundance in this fraction (6 and 5%) (Fig. 5B, left panel). Finally, some phyla were present in high abundance only in this fraction, such as Planctomycetes (Phycisphaeraceae and Pirellulaceae families) (10%) and Actinobacteria (CL500-29 marine group and hgcI clade) (12%) (Fig. 5A, left panel).

### S fraction

In the case of small colonies, the proportion of Alphaproteobacteria observed was similar to that found in the unicellular fraction (16%), their abundance was mostly explained by the Acetobacterales order (*Roseomonas* genus, 8%, Fig. 5B, left panel). On the other hand, the proportion of Gammaproteobacteria increased, with Burkholderiales order showing its highest abundance (26%) mainly due to the Comamonadaceae family (*Rhodoferax genus*, 9%), but also others families (Fig. 5B, left panel). The predominant order from Bacterioidetes was Cytophagales. Planctomycetes and Actinobacteria were less abundant in the S colonies than in the U fraction (7% and 6%, respectively) and Acidobacteria showed a relative abundance ≥ 5% only in this fraction (Fig. 5A, left panel).

### M fraction

The orders from Alphaproteobacteria showing higher abundance were Acetobacterales (especially accounted for by *Roseomonas* genus) and Caulobacterales (9% and 8%, respectively). In this size fraction Gammaproteobacteria showed a remarkable change in composition. Xanthomonadales order was detected represented by *Silanimonas* genus (5%) (Fig. 5B, left panel). Also, the composition of Burkholderiales in this fraction (20%) changed, showing a shift to Nitrosomonadaceae (6%) and Sutterellaceae (8%) families. The dominant Bacteroidetes order was Cytophagales. As observed in S, Planctomycetes (Phycisphaeraceae and Pirellulaceae families) and Actinobacteria (CL500-29 marine group and hgcI clade) were less abundant than in U (7% each) (Fig. 5A, left panel).

### L fraction

The large colonies were those showing the highest proportion of Alphaproteobacteria class (33%), while abundance of Gammaproteobacteria remained constant through S to L fraction (26–29%). The dominant members of Alphaproteobacteria were similar to those observed for the M fraction (*Roseomonas* and Caulobacterales, 8% each), with the addition of Rhizobiales order (8%). In L fraction, the Xanthomonadales order increased due to *Silanimonas* (12%, Fig. 5B, left panel), whereas the Burkholderiales order decreased its abundance (15%). As for the M fraction, the Cytophagales were the most abundant Bacteroidetes. Actinobacteria accounted for 7% (due to CL500-29 marine group and hgcI clade).

## Size-specific active bacteria and shifts between sample sizes: from presence to activity

In order to determine which bacterial taxa from the whole community were active in each sample size, the cDNA-based communities were analyzed (*Blazewicz et al., 2013*). The most active bacterial group in the whole dataset was Cyanobacteria (47–88%), being *Microcystis* the genus accounting for 35% to 86% of the reads obtained from the cDNA.

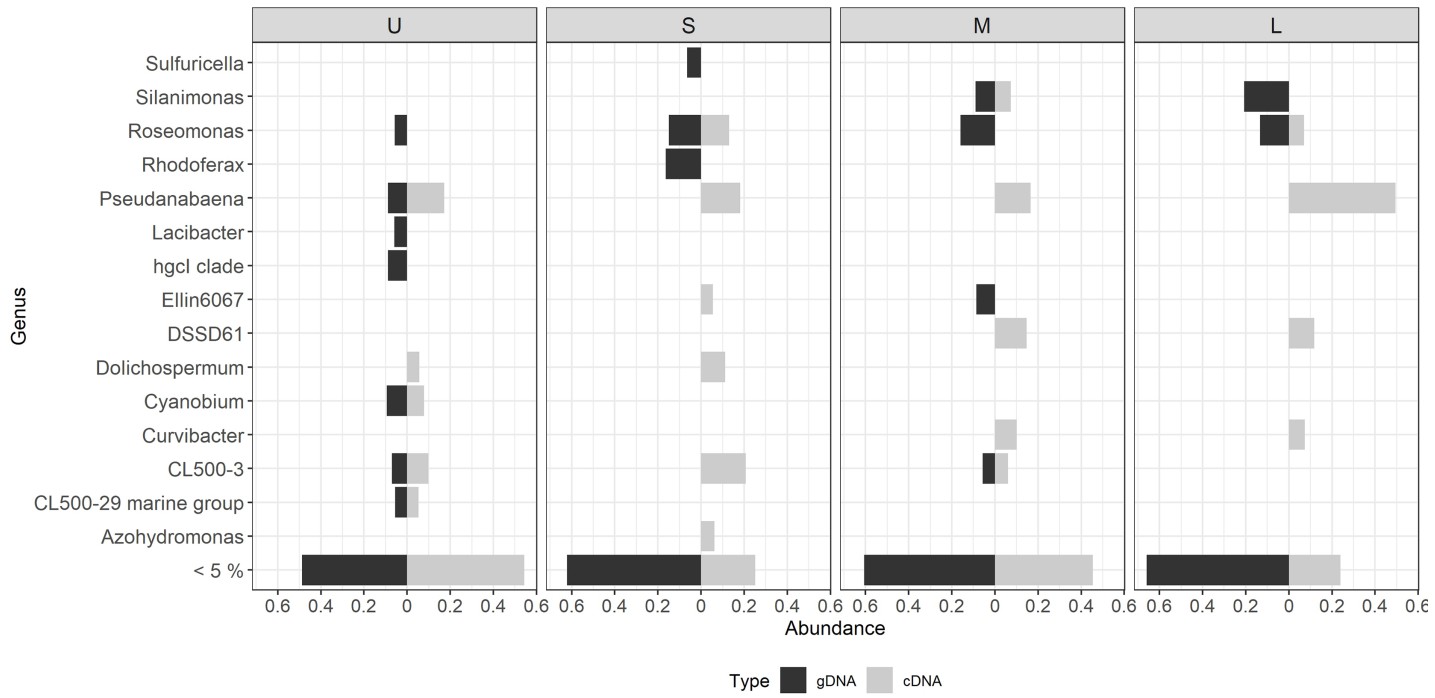

**Figure 6 Differential bacterial taxa observed between the bacterial community composition (gDNA) and the active bacterial fraction (cDNA) for each colony size.** U (Unicelular, <20 μm), S (Small, 20–60 μm), M (Medium, 60–150 μm) and L (Large, >150 μm). Only genera having >5% relative abundance are shown.

Active *Microcystis* increased along with sample size, reaching its maximum in M, and then showing a slight decrease towards L (Fig. S3). Since we wanted to analyze changes in the associated microbiome community, the *Microcystis* reads were excluded from further analysis. Proteobacteria were the highest active bacteria (28% to 52%), also showing its maximum in M (Fig. 5A). Although at the phylum level there was an agreement between the phyla having higher abundances and activity such as Proteobacteria (28% to 52%), Bacteroidetes (20% to 9%) and Planctomycetes (14% to 5%), the active taxa belonging to each phylum were different among sample sizes.

In the case of Proteobacteria, Alphaproteobacteria class showed a great differential activity among its different members, with some exceptions. The highest relative abundance of Rhizobiales was observed in U (8%), while in S the *Roseomonas* genus (Acetobacterales) was the most active one (7%) (Fig. 5B, right panel). Gammaproteobacteria also showed its highest activity in the M fraction, where Xanthomonadales (genus Silanimonas, 4%) and Pseudomonadales (genus *Pseudomonas*, 2%) appeared (Fig. 5B, right panel). In the case of Burkholderiales order, an increase of activity was observed in S (27%) compared to the U fraction (8%), mainly attributed to Sutterellaceae family (18%), whereas the activity in M (24%) and L (19%) were due to Nitrosomonadaceae (genus DSSD61), Comamonadaceae (genus *Curvibacter*) and Sutterellaceae families. In the case of Bacteroidetes, most of the orders were more abundant in U (Fig. 5A, right panel). This was mostly attributed to Chitinophagales, and

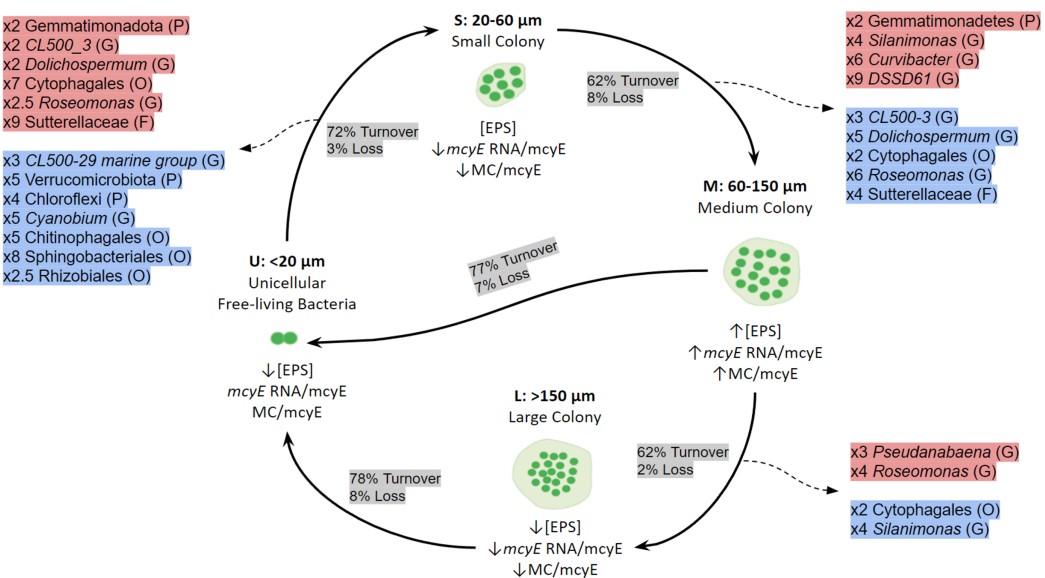

**Figure 7 Conceptual summary showing the shifts of active bacterial taxa during size transition in *Microcystis* colony development and growth.** P (Phylum), O (Order), F (Family) or G (Genus). The proportion of Turnover or Loss of species between sizes is also shown. In red, taxa showing an increase in activity between fractions; in blue, taxa showing a decrease in activity between fractions. U (Unicellular, <20 µm), S (Small, 20–60 µm), M (Medium, 60–150 µm) and L (Large, >150 µm) fractions. The x followed by a number indicates the number of times a taxon increases or decreases between fractions. Only taxa having a relative abundance ≥ 5% at the highest taxonomic rank achieved are shown.

Sphingobacteriales (5% and 8% each). In the other fractions, the most active order was Cytophagales (13% to 5%).

Regarding Cyanobacteria other than *Microcystis*, *Pseudanabaena* increased its activity towards the L fraction (from 10% in U, S and M to 30% in L fraction). Also, in the U and S fractions the genera *Cyanobium* and *Dolichospermum* became active, respectively (Fig. 5B, right panel; Fig. 6). Other phyla like Actinobacteria (due to CL500-29 marine group) (5%) and Verrucomicrobiota (5%) were only abundant in the U fraction, while Gemmatimonadota in M and L fractions (8% and 6%, respectively).

Most of the active taxa were not the most abundant when community composition based on gDNA was assessed. Examples of taxa showing a high abundance according to gDNA but low activity are the genera *Silanimonas* and *Roseomonas*, the first one reaching 12% in L but with no activity, and the second one was present with 8% in M and L but then decreasing its activity to 1 and 4%, respectively (Fig. 6). On the contrary, some genera showed the opposite pattern such as *Pseudanabaena*, which percentages were ≤ 5% in gDNA-based community but in the active community of L reached 30%, or *Curvibacter* and DSSD61, having ≤ 2% in the community but reaching an active abundance of 6% and 9% in M and L, respectively (Fig. 6).

When the mechanisms behind these shifts in community composition and in the active taxa were addressed, the dominant process was species turnover, having a higher impact on the active fraction. The Fig. 7 shows a conceptual model for colony growth and

development of *Microcystis*, based on toxin production, *mcyE* gene expression, number of cells per colony and taxa whose activity showed a remarkable increase or decrease between sample sizes.

## DISCUSSION

The role of the associated microbiome in the development of *Microcystis* blooms has recently started to be unveil. The progression from unicellular or small colonies to large, visible colonies has been systematically observed, yet little is known about the factors involved in the dynamics of colony growth. Recently, *Shi et al. (2022)* has reported that *Microcystis* and certain bacterial groups (*e.g.*, Bacteroidetes and Proteobacteria) from its associated microbiome exhibited coordinated transcriptional patterns, pointing to the existence of labor division in the colonies.

It has been shown that *M. aeruginosa* is able to produce AHLs (*Zhai et al., 2012*), and its cells not only produce transcripts involved in biofilm formation pathways, but also are able to respond to several AHLs from Gram negative bacteria by inducing colony formation, cell proliferation (or inhibition) or inducing microcystin synthesis (*Herrera & Echeverri, 2021*; *Shi et al., 2022*). This evidence suggest that colony formation may be regulated by a coordinated multicellular behavior as described for biofilms, where cyanobacteria and their associated microbiome interact to generate a complex environment that is different from the aquatic realm (*Piccini et al., 2024*). In this work, we assessed the bacterial community and its active fraction associated with different sample sizes of *Microcystis* sp. obtained from a natural bloom, which we propose that account for different phases of the biofilm (or colony) formation. The approach was based on 16S rRNA amplicon sequencing from community DNA and ribosomal RNA transcripts (cDNA) from natural bloom samples. This allowed us to obtain the bacterial community structure of the whole and active fractions associated to each sample size (*Blazewicz et al., 2013*).

### *Microcystis* associated microbiome composition and activity changed through sample sizes

The differences in the composition of the bacterial community found between different sample sizes were also observed for the active fraction, suggesting a differential activation of certain bacterial groups at each stage of colony growth. In addition, some taxa which relative abundance was high at a given sample size showed very low abundance in the active fraction, meaning that their activation would have occurred at a smaller sample size and then their growth arrested. This finding has relevance when analyzing the microbiome through metagenomic studies, as it indicates that the information provided by the community DNA does not necessarily reflect which taxa and functions are actually active.

The phylum having a higher activity was Cyanobacteria, with a main contribution of the *Microcystis* genus, which increased its activity as the sample size increased. This increase in activity that accompanied the increase in the number of cells in a colony would indicate that, at the time the bloom was sampled, they were actively growing. This makes sense since the water temperature and pH at the sampling time were in the range described as optimum for *Microcystis* growth, 27.5 °C and 9.0, respectively (*Yang et al., 2020*; *Kruk*

*et al., 2017*; *Reynolds et al., 1981*; *Wei et al., 2022*). Also, colonies in the M fraction exhibited the highest microcystin/cell concentration (Fig. 2), indicating that *Microcystis* colony growth was accompanied by toxin production. This is in agreement with findings of *Black, Yilmaz & Phlips (2011)* who showed a strong increase in microcystin concentration during growth. Also, *Orr & Jones (1998)* found that cell division rates equaled microcystin production regardless of the nutrient condition. In addition, *Long, Jones & Orr (2001)* found that under nitrogen-limited conditions, cell-specific growth rate of *Microcystis* matched microcystin production rates. These current discrepancies about which environmental cues determine the rate of microcystin production might be elucidated if biotic variables, such as the microbiome, were considered. Although a growing body of information about *Microcystis* associated microbiome exists, little is known about its relationship with microcystin production, whether the presence of microcystin is stimulated by certain members of the microbiome, or if the microcystin is a key communication molecule (*Braun & Bachofen, 2004*; *Wang et al., 2021*). As the role of the *Microcystis* associated microbiome is starting to be unveiled, further research is needed to shed light on the role of specific heterotrophic organisms in colony formation and in the persistence of *Microcystis*.

An interesting finding of the current study was that as colonies became bigger the proliferation of certain organisms occurs through selection mechanisms, as suggested by $RC_{(BC)}$. According to this, the bacterial assemblage when colonies progress from U to M sizes are mainly structured by selection probably involving biotic and/or abiotic factors present in the phycosphere, while the transition to largest sizes having the lowest bacterial richness and diversity together with low microcystin/cell concentration were more likely due to chance. Similarly, *Jankowiak & Gobler (2020)* found an inverse relationship between microbiome diversity and *Microcystis* abundance and suggested an increased selection pressure on microbial communities as blooms intensified. This agrees with our findings that suggest that during bloom onset and development, when the colonies are increasing in size, selection mechanisms probably involving signals from the cyanobacterial cells to the environment take place. On the other hand, when the bloom starts to decay the changes observed in the microbiome community are due to drift.

Although we cannot yet attribute the observed changes in microbiome composition and active taxa across the different fractions to a specific factor, we suggest that these changes could be related to variations in mucilage thickness and EPS composition. As the EPS production increases, the mucilage becomes thicker and creates a favorable environment for heterotrophic bacteria to establish (*Wang et al., 2015*; *Xie et al., 2016*). This, combined with the finding that colonies in the M fraction harbor more microcystins (*Wang et al., 2013*; *Deus et al., 2020*), could be relevant selection mechanisms to explore. In this context, the starting point of biofilm formation would begin with the attachment of selected bacterial taxa to single cyanobacterial cells (U fraction) to start the biofilm growth (analogous to S colonies) and maturation (M colonies). This process of biofilm growth and consolidation probably takes place thanks to crosstalk between certain bacterial populations from the microbiome and *Microcystis* cells. Once the biofilm reaches a larger size and architecture (L) cell disaggregation starts again (*Sauer et al., 2022*).

Analyzing samples of *Microcystis* blooms, *Yang et al. (2017)* found differences between particle-associated and free-living bacterial communities, being the particle-associated community structure conserved despite environmental changes in the surrounding water. They suggest that only those ecosystem bacteria that can adapt to the microenvironment of the colonies will live and persist in the *Microcystis* phycosphere. This would agree with *Louati et al. (2015)*, who proposed that the genus of cyanobacteria and their metabolic capabilities apparently select the bacteria that will be part of their phycosphere. If this is the case, the recruitment of bacteria from the aquatic environment to induce colony formation should depend on the functional composition of the native bacterioplankton community, the previous presence of blooms at the site, *etc.* This can be the reason why the attempts to determine the core microbiome in *Microcystis* based on taxonomic composition has failed so far (*Smith et al., 2021*; *Pérez-Carrascal et al., 2021*).

## Activation of different bacterial groups between different sample sizes

The transition from one discrete size to the next was signed by shifts in some bacterial groups. In the case of Alphaproteobacteria, the *Roseomonas* genus was associated with S, M and L fractions but showed low activity. These organisms could enrich the metabolic repertoire of *Microcystis* by encoding for complementary carotenoids pigments that are not present in the *Microcystis* genome, therefore providing an additional photoprotection and/or broaden the spectrum of light absorption (*Pérez-Carrascal et al., 2021*). They are also capable of degrading AHL (*Chen et al., 2012*), allowing to modulate the communication between *Microcystis* and other bacteria (*Chun et al., 2019*). Among Gammaproteobacteria there are different possibilities. Interestingly, while the Comamonadacea family was present and active along all fractions, Nitrosomonadaceae and Sutterellaceae were not detectable in the U fraction, being present and active in S, M and L, indicating their growth once they became part of the phycosphere. Members of the Nitrosomonadaceae family are ammonia oxidizers and it has been described that light can affect the activity of the ammonia monooxygenase enzyme. Thus, these bacteria may compete with *Microcystis* for this nitrogen source (*Podlesnaya et al., 2020*) while benefiting from the light protection provided by the mucilage. Also, some *Curvibacter* species (Comamonadacea family) are capable of providing *Microcystis* with vitamins such as B12, therefore promoting *Microcystis* growth (*Van Le et al., 2023*). Recently, *Mankiewicz-Boczek & Font-Nájera (2022)* proposed that bacteria from Sutterellaceae family can feed on *Microcystis* exudates or on decaying bloom material (*Chun et al., 2017*), probably related to the recently described ability of *Silanimonas* to degrade microcystin *via* the *mlr* gene cluster (*Yancey et al., 2023*). Here, we found active *Silanimonas* only in M fraction, which could be linked to the higher microcystin/cell content present in medium sized colonies.

Bacteroidetes showed a similar pattern to that found by *Jankowiak & Gobler (2020)*, being Sphingobacteriales and Chitinophagales the dominant orders in the free-living fraction (equivalent to U in the present study), while in the attached fraction (equivalent to S, M and L in the present study) mainly consisted of Cytophagales. The Cytophagales order is known to degrade complex macromolecules and carotenoid production (*Nakagawa, 2015*; *Cai et al., 2014*; *Lezcano et al., 2017*) and many of them have already

been identified as part of *Microcystis* microbiome, even as potential microcystin degraders (*Scherer et al., 2017*; *Lezcano et al., 2017*) or able to lyse *Microcystis* cells (*Jankowiak & Gobler, 2020*). Planctomycetes were present and active in all fractions but showed a decrease towards L. While the Phycisphaeraceae family was found to be a constant active member of the microbiome through all sizes, the Pirellulaceae family was only active in U. Similar findings were reported by *Kallscheuer et al. (2021)* who found that although the attached bacterial fraction in a bloom was dominated by members of the family Pirellulaceae, they were mostly inactive, while Phycisphaerae was the most active class within the phylum (*Kallscheuer et al., 2021*). These authors proposed that owing to their large genome size the Pirellulaceae might contain a number of catabolic enzymes required for the breakdown of the phycosphere´s polysaccharides. This would suggest that such bacteria should become active at a late bloom stage when the cyanobacterial biomass starts to decay and would explain their activity in the unicellular fraction. Organisms from Gemmatimonadota were detected in both gDNA- and cDNA- based communities but showed higher relative abundance in the latter. This phylum has members capable of accumulating intracellular phosphate and therefore may transfer phosphorus to *Microcystis* under low phosphorus conditions (*Jankowiak & Gobler, 2020*). Here, their relative abundance in the active fraction was highest at M and L, suggesting that they might play a role in providing nutrients when the *Microcystis* biomass is highest and the dissolved available nutrients start to be scarce (*Cai et al., 2024*). Actinobacteria were found across all sample sizes but were only active in the U fraction. Bacteria from this group are usually regarded as free-living (*Yang et al., 2017*; *Cai et al., 2014*) and it has been proposed that they cannot cope with the high amount of dissolved organic matter in a bloom (*Yannarell & Kent, 2009*; *Bashenkhaeva et al., 2020*). Also, some members of this phylum can utilize sunlight through actinorhodopsin, making them independent from the phytoplankton exudates (*Liu et al., 2015*; *Jankowiak & Gobler, 2020*).

We found that the *Pseudanabaena* genus was very active in the L fraction. These cyanobacteria have been described as closely associated with *Microcystis* colonies (*Sedmak & Kosi, 1997*; *Vasconcelos, Morais & Vale, 2011*; *Yarmoshenko, Kureyshevich & Yakushin, 2013*) (Fig. 1). Due to its diazotrophic behavior, it can provide *Microcystis* with fixed nitrogen (*Agha et al., 2016*; *Jankowiak & Gobler, 2020*). However, on occasions a negative effect on specific strains of *Microcystis* has been identified, causing cell lysis or buoyancy loss (*Agha et al., 2016*). Thus, *Pseudanabaena* can play a dual role, as a nitrogen provider during the last phase of colony growth and contributing to the disaggregation of large colonies and the release of single cells. In the case of *Dolichospermum*, it was only detected among the active bacteria, enriched in the U and S fractions. These diazotrophic cyanobacteria have been observed to get curled up to *Microcystis* colonies (*Jankowiak & Gobler, 2020*), where they may contribute to colony buoyancy and at the same time provide an additional nitrogen source.

## CONCLUSIONS

Previous studies about colony formation in *Microcystis* agree on the role of the associated microbiome. However, as far as we know, the studies regarding *Microcystis*-associated

microbiomes analyzed the colonies as a single entity without considering their differences in size, cell density, *etc.*, that may reflect the stages of a biofilm particle development. Here, we analyzed discrete size fractions from a *Microcystis* bloom as indicators of different stages of biofilm growth. We found differences in the microbiome community composition between fractions, which involved the activation of certain bacterial groups from one size to the next guided by turnover selection mechanisms until attain a mature biofilm (60–150 μm colony size). At this stage, microcystin concentration per cell was the highest and the enrichment of active bacterial taxa able to degrade microcystin was detected. The growth of the biofilm eventually stops when reaching colony sizes larger than 150 μm of MLD, when single cells are released to start over the cycle. This transition that we analyzed as discrete steps would occur as a continuum, where the growth of the biofilm would direct the bloom development.

The mechanisms underlying the observed transitions in bacterial community composition and the active fraction between one size and the next remain to be elucidated, but probably involve a cross-interaction between cyanobacteria and their microbiome.

## ACKNOWLEDGEMENTS

We thank Beatriz Brena and Natalia Badagian (Instituto de Higiene, Montevideo) for the ELISA analysis. This study was carried out in partial fulfillment of C. Croci requirements for the doctoral degree from Universidad de la República, Uruguay.

### Funding

The work was supported by the Agencia Nacional de Investigación e Innovación of Uruguay (ANII), grant FCE_1_2019_1_156308_20_6, and PEDECIBA-Biología. The funders had no role in study design, data collection and analysis, decision to publish, or preparation of the manuscript.

### Grant Disclosures

The following grant information was disclosed by the authors:
Agencia Nacional de Investigación e Innovación of Uruguay (ANII):
FCE_1_2019_1_156308_20_6.
PEDECIBA-Biología.

### Competing Interests

The authors declare that they have no competing interests.

### Author Contributions

- Carolina Croci performed the experiments, analyzed the data, prepared figures and/or tables, and approved the final draft.
- Gabriela Martínez de la Escalera performed the experiments, analyzed the data, authored or reviewed drafts of the article, and approved the final draft.

- Carla Kruk conceived and designed the experiments, authored or reviewed drafts of the article, and approved the final draft.
- Angel Segura analyzed the data, authored or reviewed drafts of the article, and approved the final draft.
- Susana Deus Alvarez performed the experiments, prepared figures and/or tables, and approved the final draft.
- Claudia Piccini conceived and designed the experiments, prepared figures and/or tables, authored or reviewed drafts of the article, and approved the final draft.

## DNA Deposition

The following information was supplied regarding the deposition of DNA sequences:

The sequences analyzed in this work are available at GenBank: PRJNA1169899.

## Data Availability

The code used for sequencing analysis, statistics and plots are available at:

- dada 2 package: https://benjjneb.github.io/dada2/index.html.
- phyloseq package: https://joey711.github.io/phyloseq/index.html.
- vegan package: https://github.com/vegandevs/vegan.
- betapart package: https://cran.r-project.org/web/packages/betapart/index.html.

## Supplemental Information

Supplemental information for this article can be found online at http://dx.doi.org/10.7717/peerj.19149#supplemental-information.

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
