# Peer review of "Selective enrichment of active bacterial taxa in the Microcystis associated microbiome during colony growth"

_PeerJ, doi:10.7717/peerj.19149_

## Round 0.1 · original submission · Major Revisions

I have now received comments from two reviewers. Both reviewers found your work interesting and with promising results, but also both have some concerns that should be addressed before making a final decision.

Reviewer #1 is concerned about the use of old and new taxonomy mixed in the same microbial assemblage, I recommend unifying this issue, preferentially using the modern taxonomy (as also recommended by the reviewer). Reviewer #2 has some more concerns, especially the lack of clarity in the introduction regarding the hypothesis/objective of the study and stochastically or deterministically structured microbial communities.

I recommend you follow each of the points of the reviewers and make a revised version of the manuscript.

·

Basic reporting

1) I am not so convinced of the often-used terms “Microcystis and its microbiome” or “microbiome of Microcystis”. This is because Microcystis is a microbe, too, and it doesn’t really have a microbiome like the human gut or Daphnia for example… Personally I find “Microcystis associated microbiome” or something similar more fitting.
2) Taxonomy: L 171and onward – Why did you reclassify Burkholderiales as Beta? I think Silva is now based on the revised taxonomy where Beta don’t exist anymore. From silva: “With SILVA release 138 the Genome Taxonomy Database (GTDB) has been adopted. As a consequence of our efforts the following groups were prone to significant adaptations: Archaea, Enterobacterales, Deltaproteobacteria, Firmicutes, Clostridia. Betaproteobacteriales (formerly known as Betaproteobacteria) is now Burkholderiales, an order of Gammaproteobacteria. Epsilonproteobacteria vanishes within a new phylum Campilobacterota. Tenericutes are gone, they are now all part of Bacilli, inside Firmicutes. “ (for more info see https://www.arb-silva.de/documentation/faqs/)
I think just reclassifying a single group is really wrong. If the authors don’t agree with this new taxonomy I guess they should use an old taxonomy but then completely commit to that one since there are many groups with new taxonomic names in the text. Just changing one group is a bit arbitrary.
3) I could not evaluate data sharing since the sequences are not public

small suggestions for the text:
L 116 – filtered onto ?
L 121 – for RNA later I don’t think the ® is need, maybe it would be better to mention the producer.
L 136 – Toxic cell abundance
L 153 - onward, also here I would remove the ® and TM it seems a bit arbitrarily put just for some products.
L 159 – BP not pb
L 174 – the sequencing data is not public so I couldn’t check.
L 180 & 197 - there is some problem with ( -diversity)
Fig 5 – where are the unclassified genera? Are they in the <5% or removed from the graph?

Experimental design

L 138 – Please give more detailes on qPCR, such as machine, quantification system and standard curve consruction. Also how was specificity and inhibition evaluated.
L 141 – copy number per what?

Validity of the findings

no comment

Additional comments

The authors have analysed their data in an appropriate and very interesting way attributing changes in the community structure to different ecological processes. This study was clearly well planned to analyse exactly what they did. In my opinion the study is basically flawless and the results are very interesting. Graphs are comprehensive and aesthetically pleasing. The article is well written, and of appropriate length. It was a pleasure to review this work.

Reviewer 2 ·

Basic reporting

The authors present a study examining changes in bacterial community structure and its active portion across different colony sizes (fractions) obtained during a Microcystis bloom. The work evaluates selective processes between different size fractionations. The article brings forward insight into community composition and its active proportion of taxa between 4 different size fractions and I find this article to be constructed in a relatively straightforward manner. However, the article is diminished by vague context explanation and language. The use of clear, unambiguous, professional English language could significantly improve the article's impact. My major concern with this work is the lack of clearly stated hypotheses and/or objective(s). I am uncertain of where the “working hypothesis’ (line 98) is derived from. To me, it is unclear if it set up in the introduction or uncited if it is from previous work. With this, all sections could benefit from major revisions and support from related literature.

Secondly, there is no introduction to understanding stochastically or deterministically structured microbial communities. Besides the Abstract, this concept is first mentioned in Methods (lines 195-196). As a reader, this feels like there is missing relevant background (in the Introduction) on what becomes a main discussion point of the study. As the introduction lacks relevantly cited material on microbial/bacterial colonization and growth patterns, there are several articles that would make for important Introduction and Discussion points. The findings here contradict common findings in temporal bacterial succession studies, with stochastic colonization developing into deterministic communities over time. Please see articles such as for Brislawn et al (2019), Robinson et al (2024), Wu et al (2019), for support, I’ve included the citations below. This could be of great value to discuss with size fractionation, as different dynamics are occurring between different size classes.

In terms of article structure, the structure of the article is somewhat inconsistent, with some spacing between sections/paragraphs being single or double-spaced. Please be consistent with line spacing and overall formatting.

Figures are relevant, well-formatted, and easy to comprehend, but could benefit the reader through minor revisions such as removing the grey backgrounds from plots and/or adding more distinct colours (e.g., Figure 7).

Data accessibility is present in-text.

To improve, I strongly advise the authors to clearly state the hypothesis and include more relevant references to understanding the context of the work.

Citations:
Brislawn, Colin J., et al. "Forfeiting the priority effect: turnover defines biofilm community succession." The ISME journal 13.7 (2019): 1865-1877.
Robinson, Rylie L. Investigating biofouling on long-term, in situ, optic sensors: impacts on optical measurement integrity and insight into community dynamics. Chapter 3: MS thesis. University of Windsor (Canada), 2024. Pg. 53-86
Wu, Yu-Fan, et al. "Enhanced microbial interactions and deterministic successions during anoxic decomposition of Microcystis biomass in lake sediment." Frontiers in Microbiology 10 (2019): 2474.

Experimental design

I commend the use of microscopy for size differentiation, as I know this work is exceptionally tedious and difficult. I do think this was an appropriate method suited to the study.

As mentioned, I find the research question is not well-defined. Most importantly, it is not clear upon the first read through. A secondary concern for me is addressing the research gaps this work is filling is only somewhat vaguely described in the Introduction. I do not think it is further adequately described in the Conclusion. There needs to be a tie-back to why this work is relevant in greater detail. Thirdly, rarefaction curves were generated and included in the supplementary, but it is unclear if the sequencing data was rarefied before statistical analysis. Please be clear in methods and results if it was performed, especially if there was a loss of ASVs. See Schloss (2024) for details.

Unclear if there were any sample duplicates taken or analyzed. I think for what is being presented, it is difficult to consider if there are no replicate or longitudinal (over time) sampling events. This creates some concern for the statistical power behind the PERMANOVA if the same size is n = 4 per group (e.g., cDNA and gDNA). Can you please provide more clarity on this?

Citations:
Schloss, Patrick D. "Rarefaction is currently the best approach to control for uneven sequencing effort in amplicon sequence analyses." Msphere 9.2 (2024): e00354-23.

Validity of the findings

I think the results and discussion sections are better written and clearly stated. However, the discussion section would greatly improve with more reference to previous work, as I find there is considerable points of speculation. To me, this speculation is fine if you can bring in relevant sources to support your thinking. I have added comments onto the pdf where I think this is the case.

If including environmental data, describe its impact on Microcystis development more. For example, Wei et al (2022) evaluated M.aeruginosa optimal proliferation at ~ 9.0 p. This has important implications for your system with a similar pH.

Citations:
Wei, Sijie, et al. "The proliferation rule of Microcystis aeruginosa under different initial pH conditions and its influence on the pH value of the environment." Environmental Science and Pollution Research (2022): 1-10.

Additional comments

Line 415 – 417: There is work that contrasts this statement. Comparison to this work is important to discuss because it shows a more stochastic selection process for bacteria integrating in microbial assemblages (i.e., biofilms, mats). See Robinson (2024) where plankton to assemblage communities significantly differed and Parfenova et al (2013), where taxa was similar, but composition differed. This might be of importance for Microcystis-specific communities.

Citations:

Parfenova, V. V., A. S. Gladkikh, and O. I. Belykh. "Comparative analysis of biodiversity in the planktonic and biofilm bacterial communities in Lake Baikal." Microbiology 82 (2013): 91-101.

Minor concerns: I would like to see more support for methodological choices, with consistency when citing programming packages and databases (e.g., SILVA database, {phyloseq} package). For greater replicability, I would add in manufacturer specs for materials used.

Line 180 and 197: missing symbology for either alpha or beta-diversity.

Annotated reviews are not available for download in order to protect the identity of reviewers who chose to remain anonymous.

---

## Round 0.2 · accepted · Accept

I am satisfied with the revised version; now the manuscript is accepted on PeerJ.